# A Comparative Analysis of LLM Adaptation: SFT, LoRA, and ICL in Data-Scarce Scenarios

## Abstract

The remarkable capabilities of Large Language Models (LLMs) often need to be tailored for specific applications, requiring the integration of new knowledge or the acquisition of new skills. While full fine-tuning is a powerful adaptation method, it is computationally expensive and can lead to a degradation of general reasoning abilities, a phenomenon known as catastrophic forgetting McCloskey & Cohen (1989).

A range of alternative techniques exists, each with its own trade-offs. In-Context Learning (ICL) is fast but limited by context length, while Parameter-Efficient Fine-Tuning (PEFT) methods like Low-Rank Adaptation (LoRA) offer a middle ground by minimizing parameter changes. However, the challenge of catastrophic forgetting persists, raising questions about the best adaptation strategy for a given task.

This paper presents a comparative analysis of Supervised Finetuning (SFT), LoRA, and ICL in data-scarce scenarios. We find that LoRA provides the most effective balance, successfully instilling new skills with minimal impact on the base model's general knowledge. In contrast, while SFT excels at skill acquisition, it is highly susceptible to catastrophic forgetting. ICL is effective for incorporating factual knowledge but struggles with complex skills.

Our findings offer a practical framework for selecting an LLM adaptation strategy. We highlight the critical distinction between skill acquisition and knowledge integration, clarify the trade-offs between task-specific performance and the preservation of general capabilities.

## 1 Introduction

LLMs have become foundational tools across a vast range of applications, demonstrating impressive capabilities in understanding and generating content. Despite their extensive pretraining, their utility often requires adaptation to specific knowledge or tasks. This paper provides a comparative review of these adaptation methods. However, the large number of variables, including the choice of technique, model, and associated hyperparameters, makes a complete analysis impractical. Therefore, to provide clear and actionable insights, our experiments focus on a limited set of datasets and use a single base model, Gemma 3 (Team et al., 2025).

We investigate the performance of different learning paradigms in data-scarce scenarios. The main contributions are:

1. A systematic comparison of SFT, LoRA, and In-Context Learning to benchmark their effectiveness in low-data regimes.

2. An assessment of catastrophic forgetting resulting from iterative training on small datasets.

3. An analysis of the trade-off between task accuracy and forgetting as a function of key hyperparameters such as learning rate and LoRA rank.

4. We provide a analysis of LoRA's weight updates ($\Delta W$), revealing that its stability stems from modifications to high-level layers that are established early in training.

Our experimental design intentionally starts with the standard configurations for SFT, LoRA, and ICL as provided by common implementation frameworks. Accordingly, we forgo the use of auxiliary

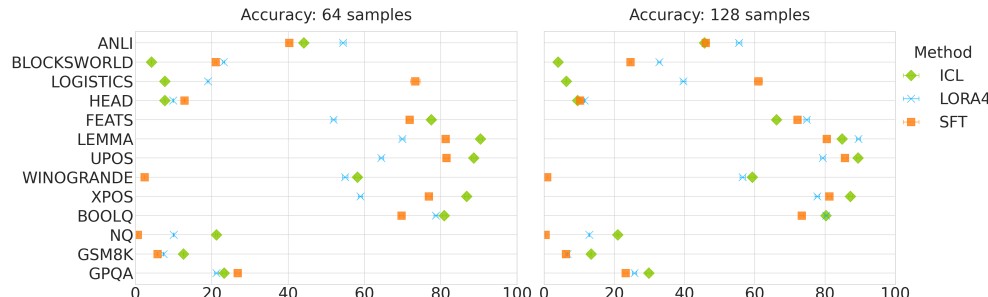

Figure 1: Per-task accuracy comparison of ICL, LoRA$_{r=4}$, and SFT across 13 benchmarks. Left: 64 samples; right: 128 samples. Performance is strongly task-dependent. On planning tasks (Blocksworld, Logistics),SFT and LoRA significantly outperform ICL. For NLP tagging skills (FEATS, LEMMA, UPOS, XPOS), ICL is best with 64 samples, while with 128 samples LoRA/SFT largely close the gap, showing only minor differences. Knowledge-heavy tasks (NQ, GSM8K, GPQA) show minimal change with more shots. Overall, moving from 64 to 128 samples generally raises accuracy for skill-based tasks.

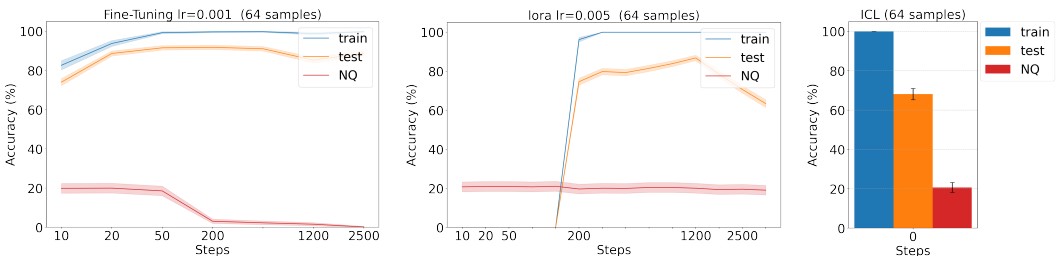

Figure 2: Skill acquisition comparison of ICL, LoRA$_{r=4}$, and SFT with 64 samples. We plot accuracy on the target skill Part-of-Speech Tagging (train/test) and a general knowledge benchmark (NQ) to measure catastrophic forgetting. Left: SFT rapidly masters the skill (high test accuracy) but suffers complete catastrophic forgetting, with NQ accuracy dropping to 0. Middle: LoRA also learns the skill effectively but, in sharp contrast to SFT, preserves general knowledge, as shown by the stable NQ accuracy. Right, ICL demonstrates partial skill acquisition at inference time (lower test accuracy) and, as it involves no weight updates, shows no knowledge degradation.

regularization techniques, such as dropout, and complex training strategies like early stopping or hyperparameter-based model selection. This methodological choice is deliberate, as our primary objective is not to achieve state-of-the-art performance on any single task. Rather the aim is to establish a controlled baseline framework to investigate fundamental questions such as how different adaptation paradigms mediate the trade-off between learning and retaining pre-existing capabilities. Each of these methods has its own profile of advantages and disadvantages. ICL is an attractive method for immediate task acquisition as it avoids catastrophic forgetting. However, for smaller, less capable models, ICL's performance may be inadequate for complex, skill-based tasks such as planning. Even with numerous in-context examples, these models can fail to master the tasks, necessitating fine-tuning techniques like Supervised Fine-Tuning or Low-Rank Adaptation to achieve the required proficiency. The curves across multiple tasks also provide a profile of the learning capabilities for a given model. While a model can learn some tasks using ICL for others ICL fails to impart the necessary skill necessitating the use of SFT or LoRA.

## 2    RELATED WORK

**ICL** is a cornerstone capability of LLMs, enabling them to perform new tasks solely from examples in a prompt, without any gradient updates (Brown et al., 2020). The efficacy of ICL was significantly enhanced by chain-of-thought (CoT) prompting, which demonstrated that including intermediate reasoning steps in the examples improves performance on complex tasks (Wei et al., 2022). However, research has predominantly focused on the "few-shot" regime, a limitation imposed by the small context windows of earlier models.

Recent architectural advances expanding context windows to over one million tokens (Gemini Team, 2024) have made the "many-shot" ICL regime newly feasible. Pioneering work in this area by Agarwal et al. (2024) shows that many-shot ICL can perform comparably to SFT on certain tasks, like low-resource machine translation. They also introduced Reinforced ICL, using self-generated rationales to achieve substantial improvements. While informed by their findings, our paper focuses on a regime with fewer "shots," does not employ the Reinforced ICL technique to compare simpler more standard approach with alternative methods. We use also a model not specifically trained to have context up to a million tokens.

**SFT** is a predominant paradigm for adapting pre-trained LLMs to specialized downstream tasks. While effective at eliciting specialized capabilities, SFT is computationally expensive and susceptible to catastrophic forgetting, a phenomenon where the model's general knowledge and performance on pre-trained tasks degrade as it specializes (Dong et al., 2024; Dou et al., 2024).

To mitigate these issues, PEFT methods have gained prominence, with Low-Rank Adaptation (LoRA) being one of the most widely adopted Hu et al. (2022). LoRA freezes the pre-trained weights and injects small, trainable low-rank matrices, drastically reducing the number of trainable parameters and the associated computational cost (Hu et al., 2022; Lee & Wynter, 2024). This efficiency has made LoRA popular for model customization (Luo et al., 2025; Shuttleworth et al., 2024).

A significant body of research posits that LoRA's constrained nature inherently mitigates catastrophic forgetting. The prevailing view, articulated in works such as Biderman et al. (2024), is that "LoRA learns less and forgets less," similar to (Lee & Wynter, 2024; Luo et al., 2025). By restricting updates to a low-rank subspace, LoRA acts as a powerful regularizer, preventing the drastic weight shifts that cause knowledge degradation in SFT (Lee & Wynter, 2024; Kalajdzievski, 2024). Empirical studies show that while SFT may achieve higher peak performance on a target domain, LoRA better preserves source-domain knowledge (Biderman et al., 2024; Lee & Wynter, 2024). This trade-off places LoRA and SFT on a learning-forgetting Pareto curve, where LoRA configurations typically occupy the region of lower target-task performance but higher knowledge retention. Consequently, LoRA is often preferred for single-task adaptation and has shown superior robustness in sequential training scenarios, exhibiting significantly less performance decay on previously learned tasks compared to SFT (Kalajdzievski, 2024).

Shuttleworth et al. (2024) argue that even when LoRA and SFT achieve identical performance on a target task, they learn structurally distinct and non-equivalent solutions, creating an "illusion of equivalence." Spectral analysis reveals that SFT induces small, diffuse changes to the model's existing weights, whereas LoRA introduces new, high-ranking singular vectors orthogonal to the pre-trained weights, which the authors term "intruder dimensions." These intruder dimensions, learned solely from the fine-tuning data, are brittle and can overpower the model's general representations and cause forgetting too.

Wistuba et al. (2023) utilize Low Rank Adaptation demonstrating that LoRA is a better choice than prompt tuning for continual learning. The paper shows that LoRA exhibits less forgetting compared to prompt-tuning.

## 3 EXPERIMENTAL SETUP AND DATASETS

Our experiments aim to identify settings where the model successfully learns new information while retaining its general abilities, such as instruction following and performing tasks unrelated to the training data. To explore this systematically, we explore hyperparameters such as the rank for LoRA or learning rate for SFT. We vary the number of training examples provided to the model on a logarithmic scale ($\log_2$) to assess the impact of amount of training data on learning and forgetting. For supervised finetuning, we further explore learning rate and training steps, to identify at which a model begins to lose its broader capabilities. We start the exploration with the standard settings (e.g. learning rate 0.001, LoRA rank 4) which is common practice and often set as standard in frameworks. We use Kauldron[1] a popular framework used with Gemma models for our experiments.

We compare the learning paradigms in data-scarce settings which crucially, permits a head-to-head comparison with in-context learning (ICL) and use up to 32k tokens with the Gemma-3 models to

---

[1]https://github.com/google-research/kauldron

accommodate the same number of examples. For a controlled comparison, we therefore run SFT and LoRA with an identical example budget and then extend the analysis to larger datasets to characterize scaling beyond this common baseline.

We divide our datasets into two categories: skill-based and knowledge-based. Skill-based tasks require the model to acquire a new capability, such as classification or playing a game (e.g., Natural Language Inference, Part-of-Speech Tagging). In contrast, knowledge-based tasks primarily evaluate providing information (e.g., Question Answering).

The following listed tasks are knowledge-orient. We also use these datasets to quantify forgetting: **BoolQ (Boolean Questions)** is a reading comprehension dataset of naturally occurring yes/no questions, each paired with a text passage from which the answer can be inferred (Clark et al., 2019). **GPQA (Graduate-Level Google-Proof QA)** dataset contains challenging, expert-level multiple-choice questions designed to be difficult even with the aid of a search (Rein et al., 2023). **GSM8K (Grade School Math 8K)** is a dataset of high-quality math problems the model has to answer (Cobbe et al., 2021). **NQ (Natural Questions)** The open-domain version of the NQ dataset, which consists of real user queries from Google Search and requires the model to generate answers from its internal knowledge without any provided context (Kwiatkowski et al., 2019).

Additionally, we use tasks that are skill-oriented, which include classification, sequence prediction, structured prediction, and planning. While the models (Gemma 3) have likely encountered these and similar tasks during pre-training, they are not typically a primary focus of the training objectives for current models. Hence they are well-suited to learn new skills and test their ability to learn those: **UPOS (Universal Part-of-Speech Tagging)** assigns a standardized grammatical category, such as noun or verb, to each word in a sentence using a consistent set of 17 universal tags applicable across multiple languages (Nivre et al., 2020). **XPOS (Part-of-Speech Tagging)** assigns labels to each word in a text with a language-specific grammatical tag that captures fine-grained morphological details like tense or case (Nivre et al., 2020). **Head (Syntactic head prediction)** is a structured prediction task that identifies the governing word (the "head") for each word in a sentence, thereby mapping the binary relationships that form a syntactic dependency tree (Nivre et al., 2020). **FEATS (morphology feature prediction)** involves assigning detailed grammatical attributes to words, such as their tense, number, or gender (Nivre et al., 2020). **LEMMA (lemma prediction)** reduces an inflected word to its canonical basic dictionary form (Nivre et al., 2020). **ANLI (Adversarial Natural Language Inference)** is a natural language inference benchmark created through an iterative, adversarial process where humans write examples specifically designed to fool state-of-the-art models (Nie et al., 2020). **Blocksworld** (Winograd, 1971) is a classic planning benchmark (e.g., IPC 2008), now used to assess the planning capabilities of LLMs (Valmeekam et al., 2023; Bohnet et al., 2024). We adopt its PDDL formulation, requiring to rearrange blocks into a goal configuration. **Logistics** is a classical planning benchmark (e.g., IPC 1998) also used to evaluate the planning capabilities of LLMs (Valmeekam et al., 2023), involves finding an optimal plan to transport packages by coordinating trucks and airplanes. **Winograd Schema Challenge (WSC)** is a coreference resolution benchmark that requires common-sense reasoning to identify the correct referent of an ambiguous pronoun in a sentence (Levesque et al., 2012).

## 4 RESULTS AND ANALYSIS

We present a comparison of ICL, LoRA, and SFT under matched training data set sizes. We report per-task accuracy across skill- and knowledge-oriented benchmarks and track retention via a held-out reference task (NQ). Beyond this baseline, we examine scaling with additional data (SFT/LoRA) and adapter rank.

**Training Settings**. For all training runs, we used a batch size of 8 to accommodate the small size of the training sets which we chose for our setup. For training, LoRA we use a learning rate of 0.005 whereas for SFT, the learning rate is listed in the experiment and between $10^{-3}$ and $10^{-4}$.

### 4.1 IN-CONTEXT LEARNING

In this section, we provide the details of evaluation of the model's In-Context Learning performance. Figure 3 shows the performance of Universal Part-of-Speech (UPOS) tagging, syntactic head predic-

tion, and Adversarial Natural Language Inference. Figure 4 shows results for Natural Questions (NQ) dataset, GPQA (Graduate-Level Google-Proof Q&A) and GSM8K.

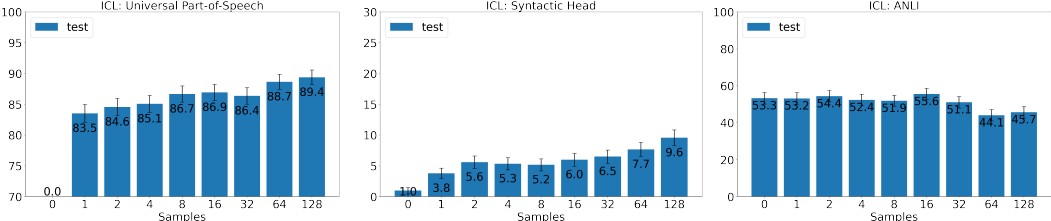

Figure 3: In-Context Learning performance on selected Skill Tasks Universal Part-of-Speech tagging, syntactic head prediction (Head), and Adversarial Natural Language Inference (ANIL). We selected two skill with improving accuracy UPOS and Head and one tasks (ANLI) with no increasing or even a drop in accuracy.

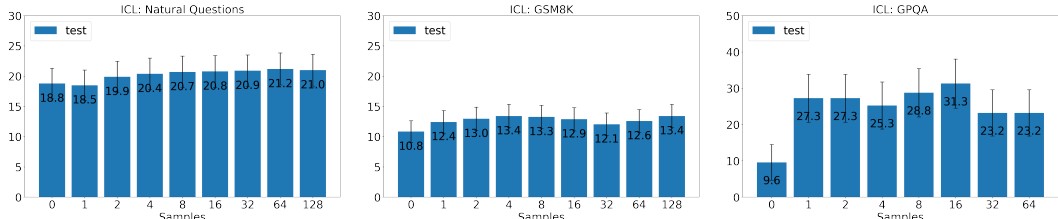

Figure 4: We evaluated the model's In-Context Learning performance on the Natural Questions, GSM8K, and GPQA datasets. In contrast to the skill task, improvements show and upward trend for Natural Questions and mixed results for GPQA.

A key finding is observed for Universal Part-of-Speech Tagging, where performance scales with the number of examples. The model's accuracy increases from 83.5% with a single example to 89.4% with 128 examples, indicating that this skill is effectively picked up by the LLM. In the middle, the results for the prediction of the syntactic head of each word in a sentence, the task is much harder and the performance remains low but increases to 9.6 % accuracy. For Adversarial Natural Language Inference, the accuracy does not change much and even surprisingly drops with more examples.

Figure 4 shows Question Answer tasks. On all three, we observed only moderate, statistically non-significant improvements. The results showed a slight positive trend with an increasing number of examples, which we attribute to the model adapting to the expected answer format rather than to substantive task learning. In contrast, accuracy on the GPQA dataset decreased after adding more than 16 samples.

## 4.2 Supervised Finetuning

SFT unlike In-Context Learning, involves updating and storing new model weights. This approach often requires multiple training epochs and risks catastrophic forgetting, where the model's general capabilities are compromised as it masters a specific task (Biderman et al., 2024).

Directly finetuning a model to embed new knowledge is known to be less effective than established retrieval-augmented methods, which are better suited for knowledge-intensive applications (Lewis et al., 2020). Consequently, we focus our experiments on skill acquisition for SFT experiments. Consistent with our previous setup, we conduct these skill-based experiments in a data-scarce setting.

Figure 5 illustrates that while fine-tuning improves accuracy on a target skill, it causes a significant degradation of general capabilities. As expected, performance on Universal Part-of-Speech (UPOS) tagging increases with more training examples. However, this specialization leads to catastrophic forgetting of unrelated skills. Specifically, the model's ability to answer questions deteriorates rapidly (see Natural Questions curve, red), and its fundamental instruction-following capacity is compromised. The model begins to erroneously annotate the instructions for other tasks instead of

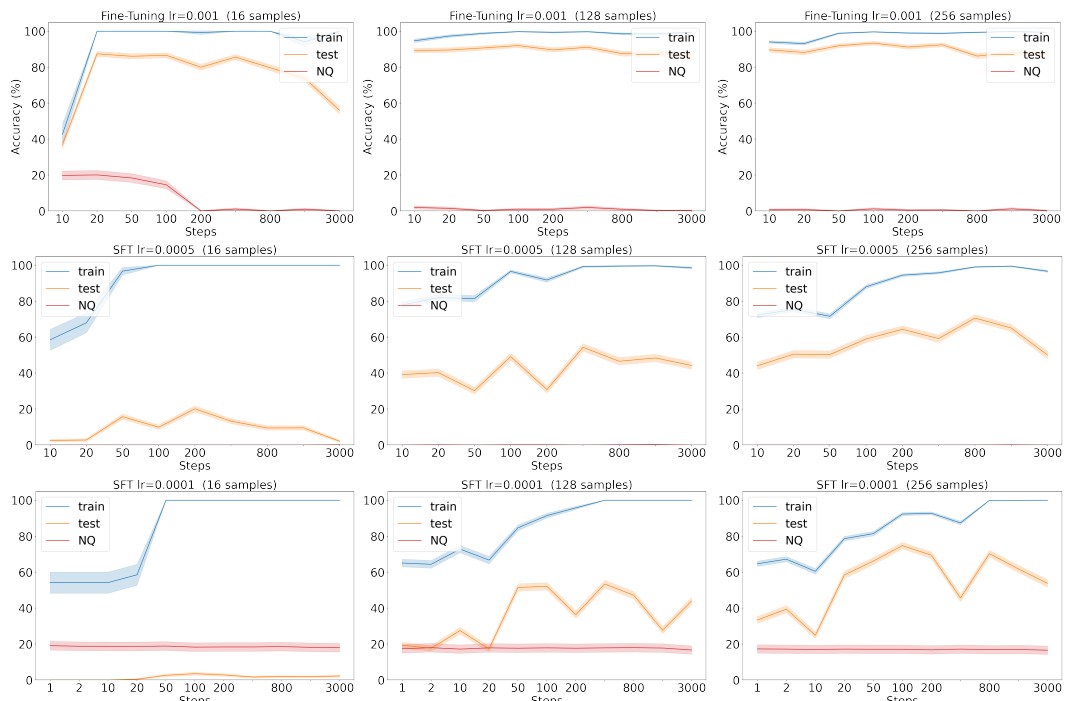

Figure 5: Supervised Fine-Tuning on UPOS results in rapid catastrophic forgetting. At learning rates of $10^{-3}$ (top), $5 \times 10^{-4}$ (middle) and $10^{-4}$ (bottom), the model's general abilities are lost within a few training steps. While the lowest learning rate ($10^{-4}$, bottom) avoids this severe forgetting, it also prevents the model from successfully acquiring the task.

executing them. This severe decay in general abilities occurs within a few training steps when using 16 to 256 training samples.

To provide a clear illustration of the learning dynamics, we present results from a single, representative training run. This approach avoids averaging or early stopping, which can obscure the fine-grained effects of catastrophic forgetting as it develops.

Furthermore, we investigated whether lowering the learning rate could mitigate catastrophic forgetting. As shown in the corresponding graphs, e.g., Figure 1, middle and bottom, this approach is **not** effective. A lower learning rate hinders the model's adaptation to the new task, resulting in poor task acquisition, while substantial forgetting of general abilities still occurs (graph in the middle). The graphs on the bottom might hint that settings exist where no catastrophic forgetting while learning a task is possible.

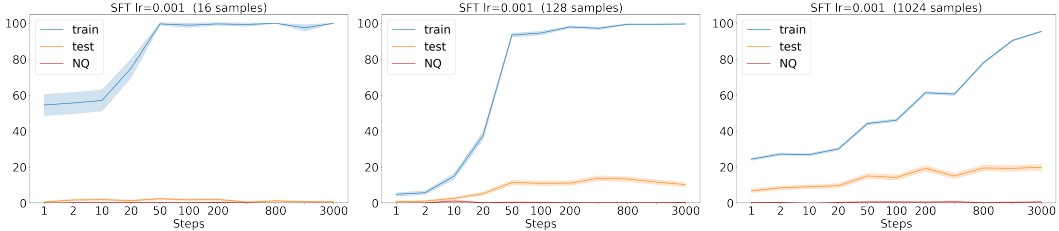

Figure 6: SFT on a structured prediction task (syntactic head identification) with a learning rate of 0.001 leads to rapid catastrophic forgetting. The model exhibits both rapid catastrophic forgetting of its general pre-trained abilities and severe overfitting on the fine-tuning data. While training accuracy quickly reaches 100% even with few samples, test accuracy scales poorly, remaining near-zero with 16 samples and reaching only approximately 20% with 1024 samples.

## 4.3 LORA

Figures 8 illustrate the data requirements of LoRA. LoRA's accuracy is shown to be dependent on training sample size, with performance improving training samples increases from 16 to 256 examples. In contrast, SFT proves more effective in the low-data regime, achieving a competitive accuracy with as few as 16 training samples as shown in Figure 5.

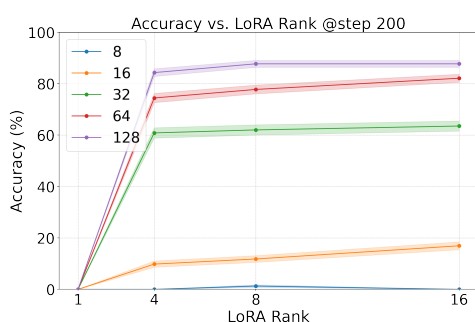

Figure 7: Test accuracy as a function of LoRA rank and number of training samples. Each model, represented by a colored line, was trained on a specific number of samples (8 to 128) for 200 steps, by which point it achieved 100% training accuracy. The plot demonstrates that generalization performance scales with both rank and data availability. We observe diminishing returns for higher ranks (e.g., r > 8 for 128 samples), indicating that both sufficient model capacity (rank) and data are necessary for effective learning.

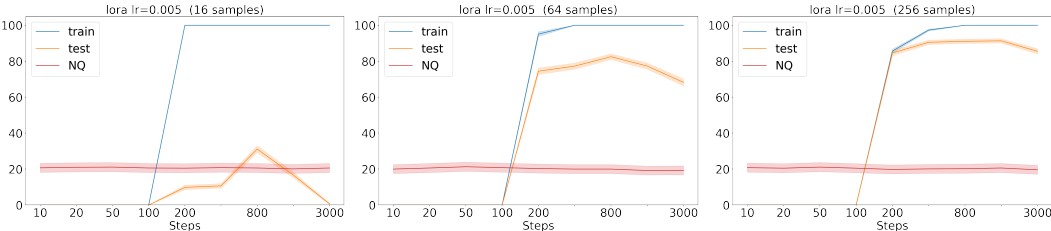

Figure 8: LoRA$_{r=4}$ with increasing number of samples. A sample size of 16 is shown to be insufficient for the model to learn effectively the task, whereas increasing the training set to 64 samples or more leads to a substantial improvement in performance.

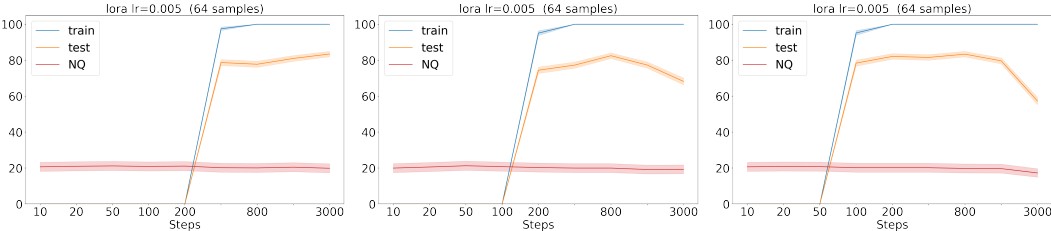

Figure 9: Effect of LoRA rank on learning dynamics. A higher rank leads to faster convergence, illustrated by the leftward shift in the learning curves. For ranks 4 and 16, peak accuracies of 82.5% and 83.3% are reached by step 800, whereas for rank 1, a peak accuracy of 83.4% is achieved with 3k steps.

Figure 10 shows that LoRA forgets too when more training samples are provided. For Part-of-Speech tagging, the accuracy continues to increase with the number of training samples, however, the ability to perform the reference task (NQ) declines. Already with 512 training samples the training accuracy of the reference task falls below the 20%. The rightmost Figure shows that the performance on the validation set reaches its peak with 8192 training samples but this comes at the cost of substantially lower accuracy on the reference task. This starts with about 400 training steps.

Figure 11 observe that the "test instructions" performance closely tracks the skill acquisition (test, orange curve) when sufficient data is provided. However, with very few samples (e.g., 8), providing instructions at test time significantly improves performance compared to evaluation on the test set alone. While this is a very promising approach to boost accuracy, the model's ability to follow these

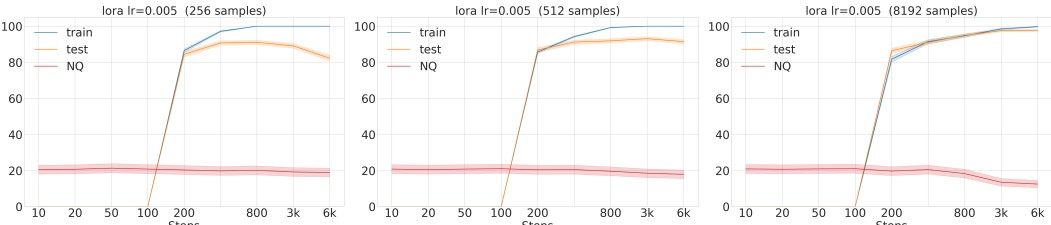

Figure 10: While Part-of-Speech tagging accuracy increases with more training samples, the reference task (NQ) declines. The performance on the validation set reaches its peak with 8192 training samples and 6k training steps, but this comes at the cost of substantially lower accuracy on the reference task.

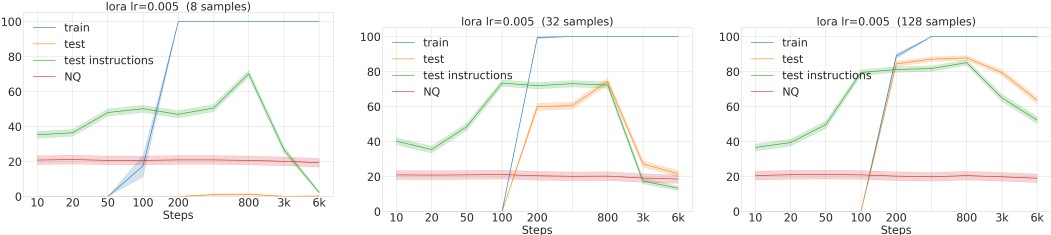

Figure 11: The impact of training samples on LoRA performance with instructions. The curves show training runs with LoRA with 8, 32 and 128 samples. The plots show accuracy for the target skill (train/test), general knowledge retention (NQ), and the model's ability to follow instructions (instructions with test set provided) on the target task (test instructions, green curve).

instructions degrades after extensive training (e.g., towards 6k steps), which may hint at a form of forgetting even with LoRA.

## 5 DISCUSSION

Our comparative analysis highlights a fundamental trade-off between learning efficiency and knowledge preservation. While SFT can rapidly acquire new skills even with minimal data, it does so at the cost of catastrophic forgetting. LoRA, in contrast, offers a more balanced solution, but its effectiveness is contingent on specific conditions regarding data availability and hyperparameter tuning. Our experiments show that LoRA fails to learn effectively from very few examples (e.g., 1 to 16 samples) under standard settings, whereas SFT excels in such scenarios (Figure 5). LoRA requires a critical mass of data to find an effective low-rank approximation for the new skill.

To understand the mechanisms behind these differences, we analyze the evolution of weight deltas ($\Delta W$) by comparing the base model's parameters to those of the fine-tuned model at various training intervals. This analysis aims to identify potential factors contributing to varying levels of catastrophic forgetting and differences in skill acquisition. Figure 12 shows the evaluation using LoRA with 32, 64, 128, and 256 training samples. The task used is Universal Part-of-Speech Tagging. **Highly Layer-Specific Updates**. The most striking feature is that the weight updates are not uniform across the model. The changes are heavily concentrated in specific layer bands. The vast majority of high-magnitude updates (green to yellow) occur in the upper-level layers, roughly from layer 20 to layer 31. The lower layers (approx. 0-8) show extremely small updates (dark blue/purple), indicating they are left largely untouched. There is a consistent, isolated band of a higher magnitude updates around layer 13 and another around layer 24. We observe that training with only few examples 32 or 64 lead mostly to adaptation in upper layers with most pronounced with 800 steps **Rapid Convergence of the Update Pattern**. The pattern of which layers are updated is established very early in the training (by 800 steps) and remains remarkably stable through to 3000 steps. The hotspots at layers 13, 24, and 28-31 are clearly visible in the 800-step plot. In the 1600 and 3000-step plots, this pattern doesn't fundamentally change. The magnitudes in some green areas (like layers 20-27) may intensify slightly between 800 and 1600 steps, but the overall distribution of where the model

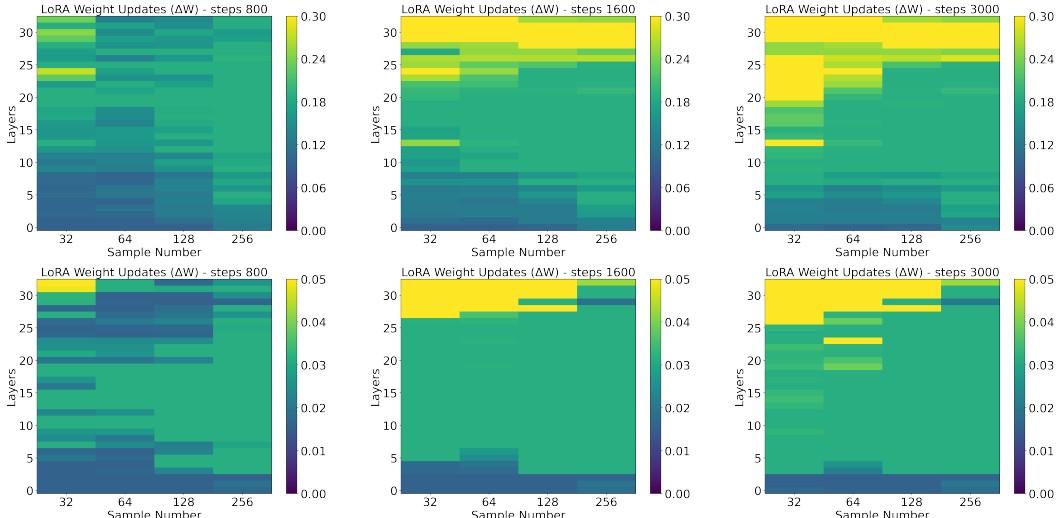

Figure 12: volution of LoRA weight update ($\Delta W$) magnitudes for attention (top row) and MLP (bottom row) weights. Each heatmap plots update magnitude across model layers (y-axis) against training samples (x-axis), with plots from left to right showing training at 800, 1600, and 3000 steps.

is being changed is fixed. **Module-Specific Updates.** Within the active layers, the updates are also not uniform across the horizontal "Sample Number" axis. For example, the hotspot at layer 13 is almost exclusively in the first column (Samples 32). Similarly, at layer 24, the largest updates are concentrated in the first two columns (Samples 32, 64).

**Hypothesis and insights**. High-level, task-specific decision rules live in late layers (Tenney et al., 2019). Low-rank updates inject changes where they most directly affect the logits, and these later layers are closest to the output. With cross-entropy loss, gradients diminish with depth during backpropagation. RMSNorm further attenuates them in early blocks, while residual connections might mitigate this in Gemma. Conversely, early layers encode reusable lexical and orthographic features that are already sufficient for the task, meaning changing them offers little gain. Attention vs. MLP roles differ; attention layers adjust routing/aggregation while MLPs adjust feature re-weighting.

## 6 CONCLUSIONS

This paper provides a rigorous comparative analysis of SFT, LoRA, and ICL for adapting Large Language Models in data-scarce environments. Our findings demonstrate that there is no single best method; instead, the choice is governed by a clear trade-off between learning efficiency, skill acquisition, and the preservation of pre-trained knowledge. SFT offers the fastest adaptation with the fewest examples but leads to rapid and severe catastrophic forgetting. ICL perfectly preserves prior knowledge but is often insufficient for teaching models complex new skills. LoRA emerges as a compelling middle ground, mitigating the worst of SFT's forgetting but requiring a larger number of training examples to become effective. For practitioners, our results provide a practical guideline: when data is extremely limited and a new skill must be taught, SFT may be necessary, but its deployment requires caution. When data is moderately available and knowledge retention is paramount, LoRA presents a more robust and scalable solution. For tasks that can be framed as demonstrations of existing capabilities, ICL remains the most efficient and safest approach.

## LLM USAGE DISCLOSURE

During the preparation of this manuscript, we utilized Gemini 2.5 Pro as a writing assistance tool to improve the quality of the text. The specific tasks performed by the LLM included identifying and correcting typographical errors, resolving syntactical mistakes, rephrasing sentences for improved clarity, and enhancing the overall coherence of the paper. All changes suggested by the LLM were critically reviewed, edited, and explicitly approved by the authors. We take full responsibility for all content presented in this work, including its scientific claims and final wording.

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
