# OpenReview forum: "A Comparative Analysis of LLM Adaptation: SFT, LoRA, and ICL in Data-Scarce Scenarios"
_ICLR.cc/2026/Conference — Submitted to ICLR 2026_

### Official Review · Reviewer_Ht2y · 2025-10-29

**Soundness:** 3
**Presentation:** 3
**Contribution:** 2
**Rating:** 2
**Confidence:** 4

**Summary:**

This paper systematically compares three LLM adaptation methods -- Supervised Fine-Tuning (SFT), Low-Rank Adaptation (LoRA), and In-Context Learning (ICL) -- under data-scarce conditions using the Gemma-3 model. It investigates how each balances new knowledge acquisition and old knowledge retention. Experiments show that SFT learns fastest but forgets prior knowledge severely, ICL fully preserves knowledge but struggles with complex skills, and LoRA strikes a balance, learning effectively with moderate data and limited forgetting, though sensitive to the sample size.

**Strengths:**

1. The experiments are comprehensive and rigorous.
2. The paper is well-written and easy to follow.

**Weaknesses:**

To be honest, the paper reads more like a technical report than a research paper. It primarily presents empirical observations and implementation details without deep investigation of new theoretical insights, algorithms, or hypotheses. As a result, while the study is informative on three learning paradigms in data-scarce scenarios, it lacks the conceptual novelty and analytical depth expected of a full conference paper.

**Questions:**

In Figure 5 (middle), catastrophic forgetting is so severe that the accuracy curve for task NQ drops close to 0 and becomes almost invisible. It would be helpful to adjust the y-axis range (e.g., –10 to 100) to improve visualization.

---

### Official Review · Reviewer_CT9Q · 2025-10-30

**Soundness:** 2
**Presentation:** 2
**Contribution:** 1
**Rating:** 4
**Confidence:** 4

**Summary:**

This paper presents a comparative study of three common LLM adaptation paradigms — Supervised Fine-Tuning (SFT), Low-Rank Adaptation (LoRA), and In-Context Learning (ICL) — in data-scarce scenarios. Using the Gemma-3 model across 13 skill- and knowledge-based benchmarks, the authors analyze performance trade-offs in learning efficiency and catastrophic forgetting.

They find that LoRA provides the best balance between skill acquisition and knowledge retention, SFT learns fastest but forgets catastrophically, and ICL preserves knowledge but struggles to acquire complex skills. The paper aims to provide practical insights for choosing adaptation strategies under low-data conditions.

**Strengths:**

The paper is well-written and clearly structured, making it easy to follow.

The comparative setup (SFT vs. LoRA vs. ICL) is useful, especially given the current interest in efficient model adaptation.

The experiments are broad (13 benchmarks across skill and knowledge tasks) and highlight consistent empirical trends.

**Weaknesses:**

* Lack of novelty. This paper summarizes known trade-offs (e.g., SFT forgets more, LoRA forgets less) without introducing new analytical insights or theoretical framing.
* The experiments cover several benchmarks, but they stay pretty descriptive. It would be nice to see more diagnostic analysis, like why LoRA fails in very low-data cases or how rank affects stability.
* The ICL results are somewhat superficial. The authors only test few-shot performance without exploring prompt design, ordering, or retrieval-based augmentation that could make the comparison fairer.
* Missing baselines: No comparison with more recent PEFT variants (e.g., AdaLoRA, DoRA, LoRA-MoE, or adapter fusion). Including one or two of these would greatly strengthen the analysis.
* Much of the cited previous work already discusses LoRA’s forgetting behavior; this paper largely re-confirms those findings rather than extending them.
* All experiments use just one base model (Gemma-3), which makes the conclusions less general.

**Questions:**

* Have you tested whether LoRA’s "upper-layer concentration" pattern (Fig. 12) also appears in other architectures or datasets?
* How sensitive are the results to LoRA’s hyperparameters?

---

### Official Review · Reviewer_nFTS · 2025-11-01

**Soundness:** 3
**Presentation:** 3
**Contribution:** 2
**Rating:** 4
**Confidence:** 3

**Summary:**

This paper compares three adaptation strategies for Large Language Models: Supervised Fine-Tuning (SFT), Low-Rank Adaptation (LoRA), and In-Context Learning (ICL) under data-scarce conditions. The study finds that while SFT achieves the fastest skill acquisition, it causes severe catastrophic forgetting, whereas ICL fully preserves prior knowledge but fails to teach complex new skills. LoRA offers a balanced middle ground, maintaining general knowledge while acquiring new abilities effectively once sufficient training data is available, providing practical guidance for choosing adaptation methods based on data and task needs.

**Strengths:**

This paper systematically investigates the forgetting problem across major LLM adaptation methods, offering extensive empirical evidence to support its conclusions. Through rigorous experiments and detailed comparisons, it provides a clear and data-driven understanding of how different techniques—SFT, LoRA, and ICL—balance learning new skills and retaining prior knowledge. The abundance of quantitative results strengthens the paper’s claims and establishes a solid empirical foundation for future research on mitigating catastrophic forgetting in LLM adaptation.

**Weaknesses:**

1. The research does not present substantially new findings beyond what is already known about catastrophic forgetting in large language models. While the empirical comparisons are thorough, the paper does not clearly explain how its results extend or challenge existing understanding from prior studies. [1,2]

2. The paper does not specify which Gemma 3 model variant was used, even though multiple versions exist (e.g., 2B, 9B, 27B). This omission makes it difficult to reproduce or contextualize the results.

[1] An Empirical Study of Catastrophic Forgetting in Large Language Models During Continual Fine-tuning

[2] How Abilities in Large Language Models are Affected by Supervised Fine-tuning Data Composition

**Questions:**

1. The catastrophic forgetting issue in LLM adaptation has been extensively studied in previous works (see weaknesses). Could the authors clarify what specific new insight or empirical finding this paper contributes beyond prior literature?
2. The paper states that experiments were conducted with “Gemma 3” but does not specify which variant was used. Could the authors provide details?

---

### Official Review · Reviewer_dT8Y · 2025-11-01

**Soundness:** 2
**Presentation:** 3
**Contribution:** 2
**Rating:** 4
**Confidence:** 3

**Summary:**

This paper presents a clear comparison of SFT, LoRA, and ICL for LLMs in data-scarce scenarios on the Gemma 3 model. The authors evaluate a series of tasks to comprehensively assess the performance of these methods. The results effectively highlight the strengths and weaknesses of each approach.

**Strengths:**

* The presentation and organization of the paper are clear and easy to follow.
* The results effectively show the strengths and weaknesses of the LLM adaptation methods on different types of tasks.

**Weaknesses:**

* The main weakness of the paper is the lack of comprehensive experiments. The experiments only consider a single model, Gemma 3, which substantially limits the generalizability of the findings. For example, the authors observe a severe performance degradation on the NQ task. Would a stronger model exhibit a similar trend? Experiments conducted on a single model make it difficult to draw convincing conclusions.
* The authors should consider including more recent and advanced variants for each method, particularly for ICL and LoRA.

**Questions:**

* How many random seeds are used to obtain the error bars in the experiments? Figure 1 appears to lack the corresponding error bar visualization. I understand that adding error bars to this figure might be challenging, but presenting them in a tabular format would also be acceptable.
* The titles of subfigures in Figure 9 seem inappropriate, the corresponding rank number is missing.

---

### Meta-Review · Area_Chair_iCH8 · 2025-12-17

**Summary:**

1. The concerns about the main contribution and novelty of this paper: This paper lacks new findings or insights beyond the previous views about catastrophic forgetting. (Reviewer dT8Y, nFTS, CT9Q, Ht2y)
2. This paper conducts experiments on one backbone (even lacks a detailed introduction), which heavily limits the generalization of the paper. (Reviewer dT8Y, nFTS, CT9Q)
3. Missing some baselines. (Reviewer dT8Y, CT9Q)

**Reviewer Concerns:**

The authors did not respond to the reviewers’ comments during the rebuttal period.

**Reviewer Scores:**

Reviewer dT8Y: retains 4

Reviewer nFTS: retains 4

Reviewer CT9Q: retains 4

Reviewer Ht2y: retains 2

---

### Decision · Program_Chairs · 2026-01-26

Reject